# Kinetic analysis of hemoglobin detergency by probability density functional method

**Miyako Oya[1], Yosuke Taniguchi[2], Naoaki Fujimura[2], Karen Miyamoto[2], Masaru Oya [2]***

**1** Surgical Department, Kimitsu Chuo Hospital, Kisarazu-shi, Chiba, Japan, **2** Graduate School of Environment and Information Sciences, Yokohama National University, Yokohama, Kanagawa, Japan

* moya@ynu.ac.jp

## Abstract

In this study, washing tests were performed using samples prepared by contaminating fabrics with hemoglobin, and a kinetic analysis was conducted based the probability density functional method, which expresses the cleaning power using two parameters $\sigma_{rl}$ (related to the cleaning mechanism) and $\mu_{rl}$ (related to the level of cleaning power). This method allows for the processing of uncertainties specific to protein washing under the assumption that the soil adhesion and detergency are in accordance with a normal distribution. A certain amount of hemoglobin solution was soaked in a cloth, dried, and steam-treated, and then used as a sample for a cleaning test. Two parameters $\sigma_{rl}$ and $\mu_{rl}$ were calculated based on the detergency (%) after 5 min, 10 min, 15 min, and 20 min of washing with respect to different pH and temperature levels, and different sodium dodecyl sulfate (SDS) concentration and temperature levels. Based on the results, the value of $\sigma_{rl}$ indicated that the hemoglobin was removed by the dissolving action. In addition, $\mu_{rl}$ increased in accordance with an increase in the pH, SDS concentration, and temperature. With respect to $\mu_{rl}$, the relationship of $\Delta X + \Delta Y = \Delta(X+Y)$ was observed in several cases, where $\Delta X$ represents the effect of the pH or SDS concentration, $\Delta Y$ is the temperature effect, and $\Delta(X+Y)$ is the combined effect. Therefore, there may be an additive relationship between the pH and temperature effects, and the SDS concentration and temperature effects.

## Introduction

In the medical field, with respect to hygiene, the cleaning of blood stains adhered to medical devices and surgical material is critical. Therefore, significant research has been conducted thereon. For example, in several studies, the effectiveness of automatic cleaning was compared with that of manual cleaning for the cleaning of medical instruments such as special cutting instruments, forceps, and endoscopes [1–4]. Moreover, studies were conducted on the effects of cleaning agents and disinfectants on the cleaning of medical devices and materials [5–7]. In several studies, adenosine triphosphate (ATP) tests were conducted, and alternative practical detergency evaluation methods were implemented [8–13], which highlighted the importance of monitoring [14–16]. Several fundamental studies were conducted with focus on the interaction of blood proteins such as albumin and hemoglobin with surfactants [17–19].

**Data Availability Statement:** All relevant data are within the paper and its Supporting Information files.

**Funding:** The author (Masaru Oya) disclosed receipt of the following financial support for the

research, authorship, and/or publication of this article: This work was supported by Grant-in-Aid for Scientific Research (B) (17H01953) and Grant-in-Aid for Scientific Research (A) (17H00814) from Japan Society for the Promotion of Science (JSPS).

**Competing interests:** The authors have declared that no competing interests exist.

Moreover, given that the theoretical analysis of the cleaning phenomenon requires an index of dirt removal power (i.e., cleaning power), the application of the rate constant of the first-order reaction equation of chemical reaction kinetics has attracted significant research attention [20–24]. Recent kinetic studies conducted on protein removal were mostly related to cleaning in the food industry. For example, there were studies that included the evaluation of the detergency of whey protein from the initial rate constant of the first-order reaction equation [25] and the determination of the cleaning power from the initial gradient of gelatin cleaning using a quartz crystal microbalance (QCM) [26]. In addition, discussions were presented on the detergency of proteins attached to stainless-steel balls from the initial desorption rate [27], and the effects of external electric fields on the removal of proteins by enzymes were clarified based on the evaluation of the initial rate [28]. As mentioned above, in many kinetic studies conducted on protein washing, the initial gradient in the first-order reaction approximation was used as an index. However, the majority of practical cleaning data is difficult to represent using a single first-order reaction equation.

In general, the initial slope of the time-dependent removal curve tends to be larger than the slope of the theoretical curve, and the final slope tends to be smaller than the theoretical curve. Therefore, the assumption of the cleaning process in the secondary chemical reaction [29] and the analysis method for the cleaning process as the sum of the two first-order chemical reactions [30–32] were proposed. However, it is difficult to give logical meaning in relation to the cleaning mechanism with methods such as the cleaning rate being proportional to the square of the residual soil amount or dividing the soil into two types.

Analysis using a simple first-order reaction equation is difficult in that when the same dirt is used, the ease of removal varies depending on the state of the attachment sites and bonding force. A potential method for the mitigation of this variation is the use of a probability density function, which is the basis of probabilistic risk analysis, as was applied in many studies; i.e., the calculation of the probability of structural collapse [33] and the ecological risk assessment of chemicals [34].

An attempt was made to apply the probability density function to the detergency evaluation. The washing results of an oily dye stain [35], iron oxide particle dirt [36], and commercially available washing test cloth soiled with a mixed stain [37] were analyzed with respect to two parameters of the probability density functional method, the mean $\mu_{rl}$ and standard deviation $\sigma_{rl}$ of the removal load distribution, assuming that the cleaning power follows a normal distribution. It was recently demonstrated that one of the two parameters ($\sigma_{rl}$) is related to the soil type [38] and the soil removal mechanism [39].

In this study, washing tests were conducted using samples prepared by the soiling of fabric substrates with hemoglobin, which is one of the main components of blood stain; and the effects of the pH, temperature, and surfactant concentration were examined based on the probability density functional method focusing on the parameter $\mu_{rl}$. Since an alkaline solution is often used as a detergent for protein stains, it can be expected that the washing rate will change significantly depending on pH. Further, since SDS has a hemolytic action, it is expected that the concentration of SDS greatly contributes to the removability of hemoglobin stains. Moreover, since both can be regarded as a kind of chemical reaction, it can be predicted that they are affected by temperature. Therefore, to confirm the additive effect under different cleaning conditions, the summation rules were investigated between the pH and temperature effects, and between the surfactant concentration and temperature effects using the parameter $\mu_{rl}$.

## Materials and methods

### Material preparation

For the preparation of the artificial soiled cloth, cotton canequim (Laundry Science Association in Japan (weave density: 22.5/10 mm warp, 24.0/10mm weft, weight: 15.2 mg/cm$^2$) and hemoglobin (derived from bovine blood, Fujifilm Wako Chemicals) were used as protein soil models. Sodium hydroxide (Fujifilm Wako Chemicals, guaranteed grade) was used for the pH adjustment (temperature was controlled in the washing machine) of the cleaning liquid and sodium dodecyl sulfate (SDS, Fujifilm Wako Chemicals) was used as a surfactant.

Approximately 250 g of cotton canequin cloth was immersed in 2 L of 0.5% aqueous sodium carbonate solution at 60–70˚C. The cloth was then rinsed using distilled water, dehydrated, air-dried, and cut it into sections with dimensions of 5 × 5 cm$^2$. A soil solution was prepared by dissolving 2 g of hemoglobin into 100 mL of 0.1 N aqueous ammonia solution, and 400 μL was added dropwise to each section of cotton cloth. Immediately after the soiling, dry heat treatment was conducted at 150˚C for 2 min in a dryer (DRD 320A, Toyo Engineering). Thereafter, the samples were steamed for 8 min in a microwave oven (HEALSIO AX-2000, Sharp) with a steam generation function, and then stored in an incubator (BITEC-300, Shimadzu) at 20˚C for 24 h prior to the cleaning test.

A tergotometer (Daiei Kagaku, TM-4) was used for the cleaning test, and the cleaning conditions were as follows. Five samples of test cloth were washed using 1 L of cleaning solution at a stirring speed of 80 rpm, and the cleaning time was 5 min per unit. The detergency (D%) was calculated from the reflectance measured using a digital colorimetric color difference meter (ZE-2000, Nippon Denshoku Industries Co., Ltd.). The surface reflectance was measured by overlaying five sheets of soiled cloth with similar colors. The K/S value was calculated using the Kubelka–Munk equation (Eq 1), with the average value obtained by measuring the surface reflectance (Y value) at the front and back of the sample as the R value, and the cleaning rate was calculated using Eq 2:

$$K\Big/S = \frac{(1-R)^2}{2R} \tag{1}$$

$$D(\%) = \frac{K/S_{\mathrm{s}} - K/S_{\mathrm{w}}}{K/S_{\mathrm{s}} - K/S_{\mathrm{o}}} \tag{2}$$

where K is the light absorption coefficient, S is the light diffusion coefficient, D is the cleaning efficiency, $K/S_S$ is the K/S value of the soiled cloth, $K/S_W$ is the K/S value of the washed cloth, and $K/S_0$ is K/S value of the raw white cloth. The K/S detergency (%) of the soiled fabric prepared using this method was found to be consistent with the detergency (%) obtained using the Lowry method after extraction using 0.1 N NaOH aqueous solution, with the exception of washing using bleaching agents [40, 41].

### Methodology

In the probability density functional method [35–39], the adhesion force of soil and the removal force of soil are assumed to be in accordance with a normal distribution, and the cleaning efficiency is determined based on both distributions. Further assumptions are that the adhesion force of soil is in accordance with a normal distribution with an average value of 0 and a standard deviation of 1 (Eq 3), and the cleaning power is in accordance with a normal

distribution with an average value of $\mu_{rl}$ and a standard deviation of $\sigma_{rl}$ (Eq 4).

$$f_{A_{n=0}}(x) = \frac{1}{\sqrt{2\pi}} exp\left(-\frac{x^2}{2}\right) \tag{3}$$

$$g(x) = \frac{1}{\sqrt{2\pi\sigma_{rl}^2}} exp\left(-\frac{(x - \mu_{rl})^2}{2\pi\sigma_{rl}^2}\right) \tag{4}$$

where $n$ is the number of cleaning cycles, $x$ is a random variable, and $\pi$ is the circumference. To calculate the removal amount, the removal load $\varphi(x)$ obtained by subtracting the cumulative function of $g(x)$ from 1 is calculated (Eq 5):

$$\varphi(x) = 1 - \int_{-\infty}^{x} g(t)dt \tag{5}$$

The calculation of the product of $\varphi(x)$ and the adhesion distribution of soil yields the amount of removed dirt; which can be expressed by Eq 6 when $n = 1$, and by Eq 7 when $n > 1$.

$$f_{R_1}(x) = f_{A_0}(x) \times \varphi(x) \tag{6}$$

$$f_{R_n}(x) = f_{A_{n-1}}(x) \times \varphi(x) \tag{7}$$

The detergency (%) is related to changes in the area of the adhesive force distribution peak. The ratio of the soil removal amount ($R_n$) due to $n$ wash cycles can be calculated by the integration of $f_{Rn}(x)$ (Eq 8), and the removal efficiency ($D_n$) due to $n$ wash cycles can expressed by Eq 9. It should be noted that the integral value of $A_0(x) = 1$.

$$R_n = \int_{-\infty}^{\infty} f_{R_n}(x)\,dx \tag{8}$$

$$D_n = D_{n-1} + \left(\frac{R_n}{\int_{-\infty}^{\infty} f_{A_0}(x)\,dx}\right) \times 100 \tag{9}$$

Moreover, the distribution of residual dirt can be obtained by subtracting the distribution of the removed parts from the distribution before cleaning. When $n = 1$ and $n > 1$, the distribution can be expressed by Eq 10 and Eq 11, respectively.

$$f_{A_n}(x) = f_0(x) - f_{R_n}(x) \tag{10}$$

$$f_{A_n}(x) = f_{A_{n-1}}(x) - f_{R_{n-1}}(x) \tag{11}$$

Based on this theory, upon the determination of $\sigma_{rl}$ and $\mu_{rl}$, the distribution of the removed dirt and residual dirt can be predicted, and the amount of removed dirt and the detergency (%) can be calculated. In this study, the mean value ($\mu_{rl}$) and standard deviation ($\sigma_{rl}$) of the removal force distribution were calculated by comparing the experimental detergency (%) with the calculated detergency (%) with respect to different values of $\mu_{rl}$ and $\sigma_{rl}$. The optimal values of $\mu_{rl}$ and $\sigma_{rl}$ were then determined using the least squares method. It should be noted that the calculation ranges of $\mu_{rl}$ and $\sigma_{rl}$ were –10.0–10.0 and 0.01–10.00, respectively.

The relationship between the $\mu_{rl}$ and $\sigma_{rl}$ values and the time-dependent cleaning rate curve is shown in **Fig 1**. With an increase in $\mu_{rl}$, the detergency distribution curve shifts to the right, and the detergency curve shifts upward over the entire range. Moreover, with an increase in $\sigma_{rl}$, the width of the cleaning power distribution increases, and the height decreases. In

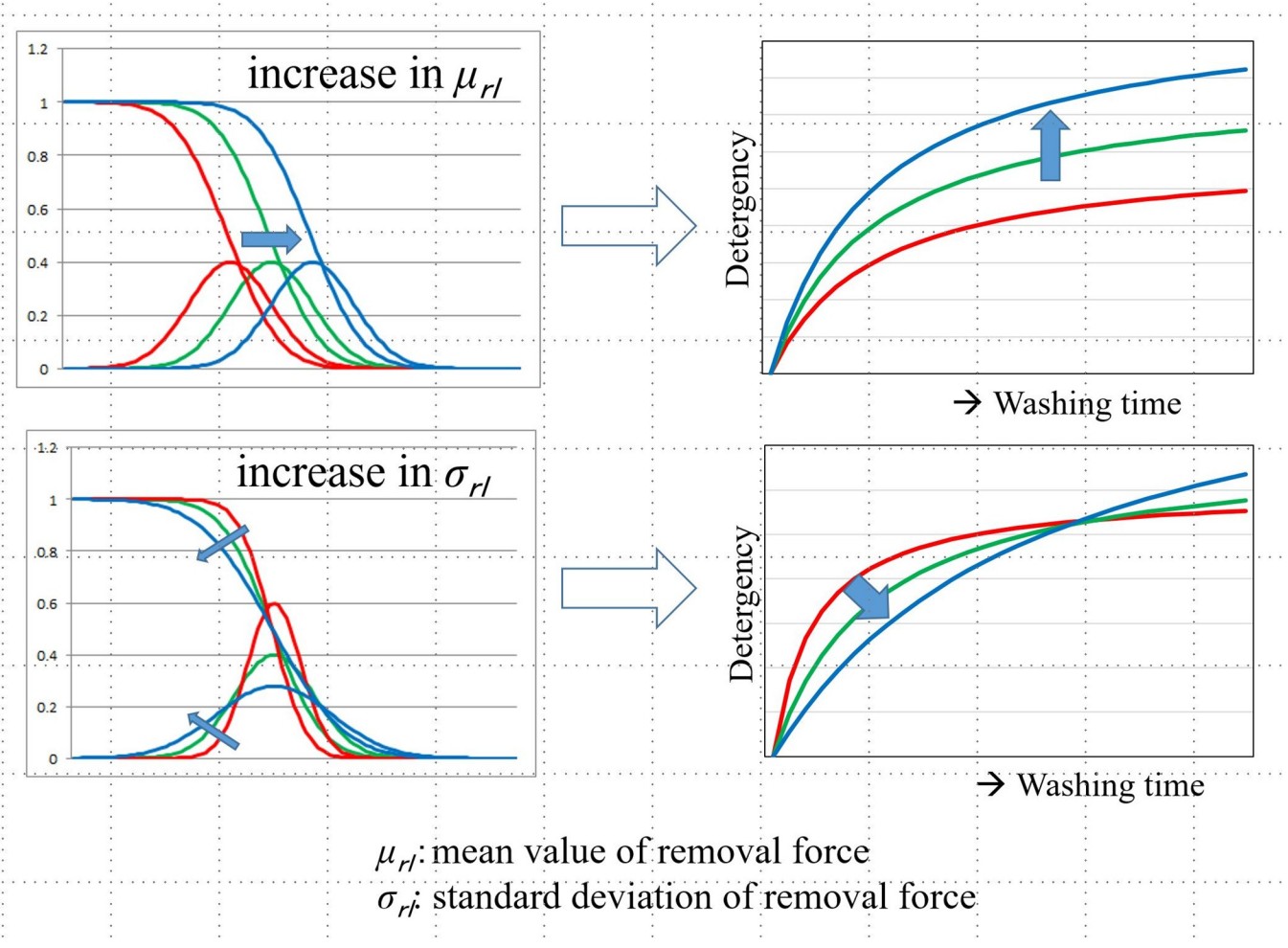

**Fig 1. The meaning of two parameters $\mu_{rl}$ and $\sigma_{rl}$.**

addition, the cleaning rate curve exhibits a larger gradient at the initial stages and a smaller gradient at the final stages.

## Results and discussion

### Washing efficiency with respect to variations in the pH and temperature

The time-dependent curves of the detergency obtained by varying the pH in steps of 6, 9, 10, and 11 at cleaning temperatures of 30˚C, 40˚C, 50˚C, and 60˚C, respectively, are shown in **Fig 2**.Since it was expected that the cleaning efficiency would be low when the pH was low at low temperature and the cleaning effect due to the temperature and pH increase would be difficult to appear, the minimum temperature was set to 30˚C. Under all the temperature conditions, the cleaning efficiency continuously increased in accordance with an increase in the cleaning time, and the detergency was found to increase in accordance with an increase in the temperature and pH.

Based on the analysis of the washing test results with respect to the probability density function method, $\sigma_{rl}$ in the range of 0.39–0.70 was obtained (**Table 1**). Moreover, a direct relationship was found between $\sigma_{rl}$ and the cleaning mechanism, and the cleaning mechanism varied

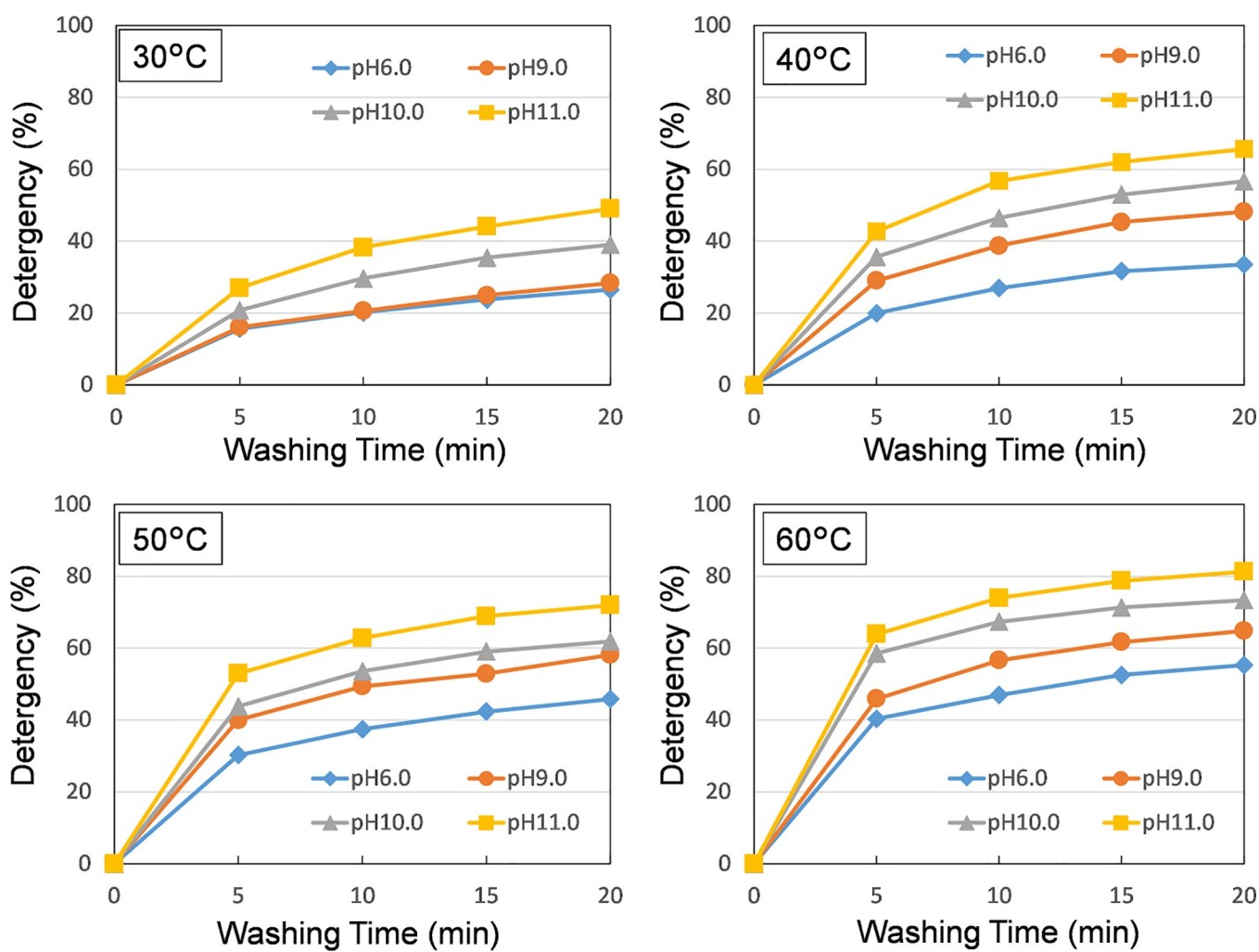

**Fig 2. Time dependent detergency curves of hemoglobin soil obtained with respect to various temperature and pH levels (soiled fabrics were washed, and the pH was controlled using NaOH).**

in accordance with an increase in $\sigma_{rl}$ in the following order: the detachment of the solid particle soil ($\sigma_{rl}$: 0.01–0.6), the dissolution of water-soluble soil (limited to substances that have a chemical binding force with fiber) ($\sigma_{rl}$: 0.3–1.4), the removal by solubilization of oily soil with aqueous surfactant solution ($\sigma_{rl}$: 1.0–2.0), the emulsification of oily soil with aqueous surfactant solution, and the removal by oily substance dispersion ($\sigma_{rl}$: 3.0-) (**Fig 3**) [39]. Given that the experimental value of $\sigma_{rl}$ was in the range of dissolution of the water-soluble soil, the hemoglobin was assumed to be removed by dissolution.

**Table 1. Calculated $\sigma_{rl}$ and $\mu_{rl}$ obtained from cleaning test of hemoglobin-soiled cloth with respect to various temperature and pH levels.**

| pH | 30°C | | 40°C | | 50°C | | 60°C | |
|---|---|---|---|---|---|---|---|---|
| | $\sigma_{rl}$ | $\mu_{rl}$ | $\sigma_{rl}$ | $\mu_{rl}$ | $\sigma_{rl}$ | $\mu_{rl}$ | $\sigma_{rl}$ | $\mu_{rl}$ |
| 6.0 | 0.44 | -1.12 | 0.49 | -0.94 | 0.44 | -0.58 | 0.39 | -0.28 |
| 9.0 | 0.50 | -1.14 | 0.58 | -0.64 | 0.47 | -0.29 | 0.51 | -0.12 |
| 10.0 | 0.66 | -0.98 | 0.59 | -0.43 | 0.49 | -0.18 | 0.41 | +0.23 |
| 11.0 | 0.70 | -0.75 | 0.65 | -0.21 | 0.53 | +0.07 | 0.53 | +0.40 |

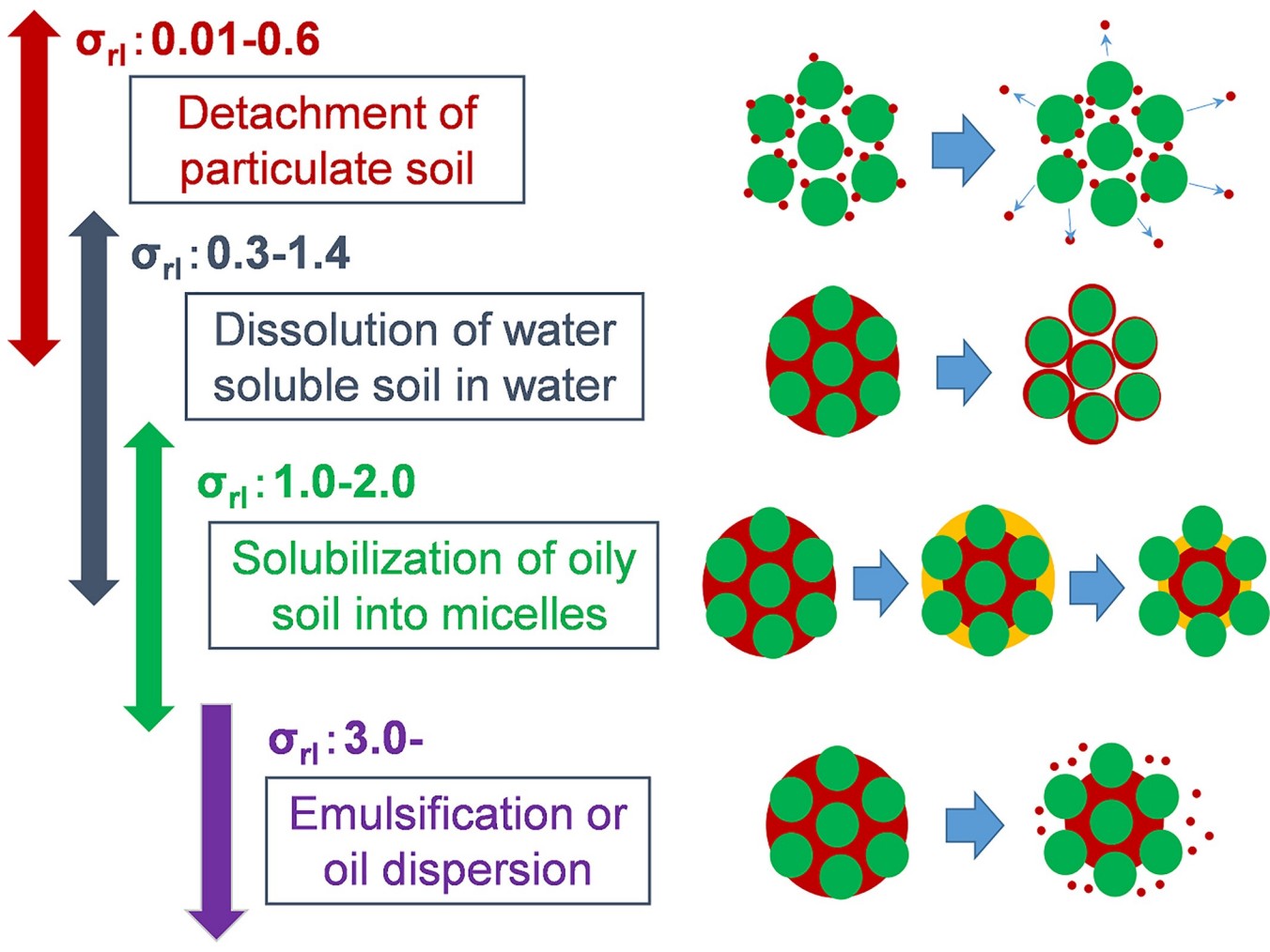

**Fig 3. Relationship between the soil removal mechanism and values of $\sigma_{rl}$.**

The value of $\sigma_{rl}$ was found to increase slightly in accordance with an increase in the pH. Although hemoglobin stains are essentially removed by dissolution, the mechanism by which solid stains are mechanically removed is more effective at low pH levels, and pure dissolution is more effective at high pH levels. However, the influence of the variation in $\sigma_{rl}$ was slight, given that no significant difference was observed when $\mu_{rl}$ was calculated with a constant $\sigma_{rl}$ at an average value of 5.2.

The dimensions of $\mu_{rl}$ are the same as those of the dirt adhesion force distribution. In addition, the dirt adhesion force distribution generally corresponds to a logarithmic plot, i.e., an ln $k$ scale where $k$ is a general rate constant. The plot of $\mu_{rl}$ with respect to the horizontal scale of $1/T$, which is the reciprocal of the absolute temperature, yielded a nearly linear relationship (Fig 4). Therefore, it was found that $\mu_{rl}$ can be adopted as a significant thermodynamic parameter.

## Verification of summation rule of pH and temperature effects

In this study, the increase in the pH and temperature resulted in an increase the cleaning efficiency under the experimental conditions. By expressing the effect of increasing the pH as

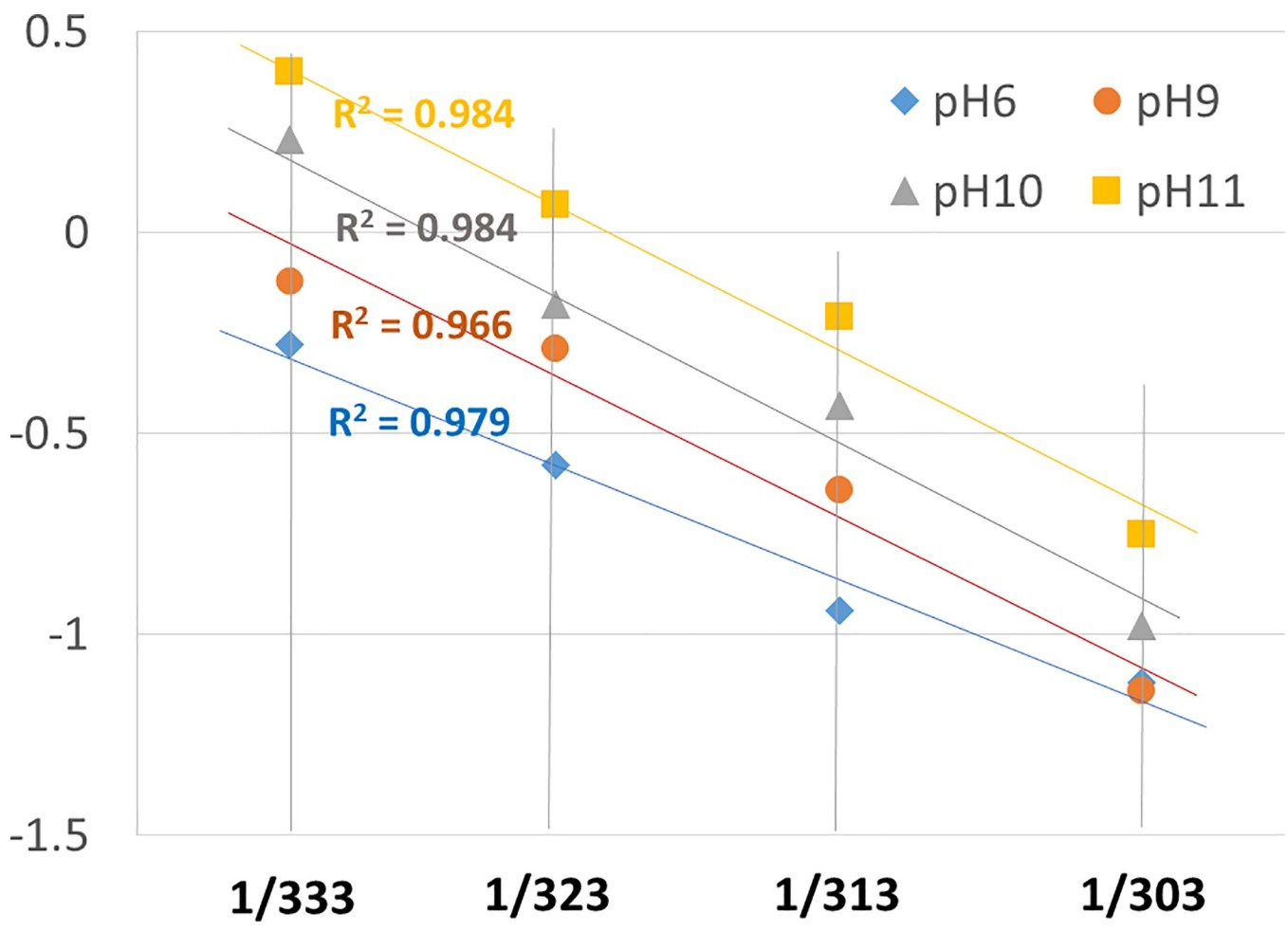

**Fig 4. The relationship between $\mu_{rl}$ and $1/T$ obtained from the washing test with respect to various pH and temperature levels.**

$\Delta D_{(pH)}$, the effect of increasing the temperature as $\Delta D_{(TMP)}$, and the effect of increasing the pH and temperature as $\Delta D_{(pH + TMP)}$, the following relationships can be obtained:

Additive effect: $D_{(pH + TMP)} = \Delta D_{(pH)} + \Delta D_{(TMP)}$

Synergistic effect: $D_{(pH + TMP)} > \Delta D_{(pH)} + \Delta D_{(TMP)}$

Offsetting effect: $D_{(pH + TMP)} < \Delta D_{(pH)} + \Delta D_{(TMP)}$

Therefore, the establishment of the additive action for nine conditions under which the pH and temperature were changed was examined, as shown in **Fig 5**. The detergency is represented by $\mu_{rl}$, and an increase in detergency is represented by an increase in $\mu_{rl}$. For example, if $\mu_{rl}$ at pH 7 is expressed as $\mu_{rl(pH = 7)}$ and $\mu_{rl}$ at pH 10 is expressed as $\mu_{rl(pH = 10)}$; $\Delta D_{(pH)}$, which is the effect of increasing the pH from 7 to 10, can be expressed as $\mu_{rl(pH = 10)} - \mu_{rl\,(pH = 7)}$. Similarly, under the assumption that $\mu_{rl}$ at a temperature of 30°C is $\mu_{rl(30°C)}$ and $\mu_{rl}$ at a temperature of 40°C is $\mu_{rl(40°C)}$; $\Delta D_{(TMP)}$, which is the increase in cleaning power when the temperature is increased from 30°C to 40°C, can be expressed as $\mu_{rl\,(40°C)} - \mu_{rl\,(30°C)}$. Moreover, $\Delta D_{(pH + TMP)}$ represents an increase in $\mu_{rl}$ when the temperature and pH are increased from 30°C to 40°C and from 7 to 10, respectively. A novel method was developed in this study to determine the additive effect by comparing the values of $\Delta D_{(pH + TMP)}$ and $\Delta D_{(pH)} + \Delta D_{(TMP)}$.

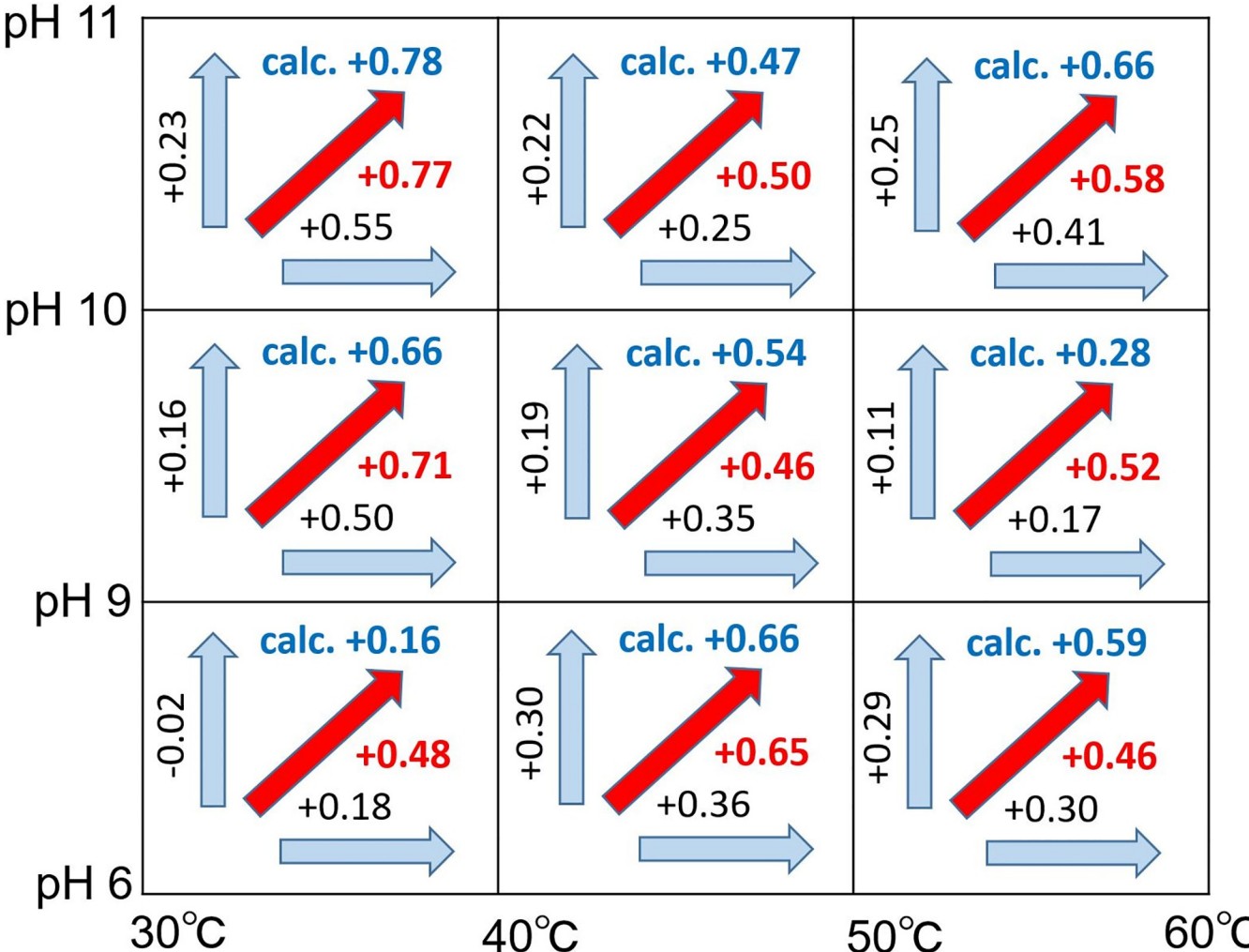

**Fig 5. Determination of additive effect between the pH increase and temperature increase in hemoglobin cleaning: The increase in $\mu_{rl}$ due to an increase in pH is expressed as a vertical value, and the increase in $\mu_r$ due to an increase in temperature is expressed as a horizontal value.** In addition, the increase in $\mu_{rl}$ due to an increase in the pH and temperature is expressed as a diagonal arrow. Moreover, additivity is confirmed if the calculated value (vertical value + horizontal value) is significantly close to the corresponding experimental value (value of red diagonal arrow).

Significant additive effects were observed under the conditions of [pH 6, 40°C → pH 9, 50°C], [pH 9, 30°C → pH 10, 40°C], [pH 10, 30°C → pH 11, 40°C], and [pH 10, 40°C → pH 11, 50°C]. In addition, relatively significant additive effects were observed under the conditions of [pH 6, 50°C → pH 9, 60°C], [pH 9, 40°C → pH 10, 50°C], and [pH 10, 50°C → pH 11, 60°C].

Moreover, under the conditions of [pH 6, 30°C → pH 9, 40°C] and [pH 9, 50°C → pH 10, 60°C], the calculated values were significantly lower than the measured values. The cleaning efficiency is dependent on complex factors such as the state of soil adhesion and uneven cleaning conditions, which results in variations. Hence, it is difficult to obtain theoretical calculation results with adequate consistency. However, it should be noted that the addition rule was satisfied under seven of the nine conditions. Furthermore, the addition rule may be established between the pH effect and the temperature effect during the dissolving action of hemoglobin in water.

## Effects of surfactant concentration and temperature

**Fig 6** presents the time-dependent curve of the cleaning efficiency when the SDS concentration was 0 g/L, 1.5 g/L, 3.0 g/L, and 5.0 g/L and the temperature was 20°C, 35°C, 50°C, and 65°C, respectively. Given that the soiled cloth from the previous experiment was used in this experiment, the absolute value of the detergency (%) cannot be compared with those of the previous experiment, as shown in **Fig 2**. The detergency increased when the surfactant concentration increased in the order of 0 g/L → 0.15 g/L → 0.30 g/L; however, no significant changes were observed between 0.30 g/L and 0.50 g/L. Given that the critical micellar concentration (cmc) of the SDS used in this study was approximately 0.25 g/L, a significant difference in the removability of the surfactant was not observed when it exceeded the cmc.

Based on the probability density function method, the results shown in **Table 2** were obtained. The range of temperature change was larger than that in the pH experiment (Table 1), and $\sigma_{rl}$ was significantly large at 20°C. This indicates that the dissolution by water and the solubilization effect by the surfactant was significant under low temperatures. At a temperature of 35°C or higher, $\sigma_{rl}$ was estimated to slightly increase in accordance with an

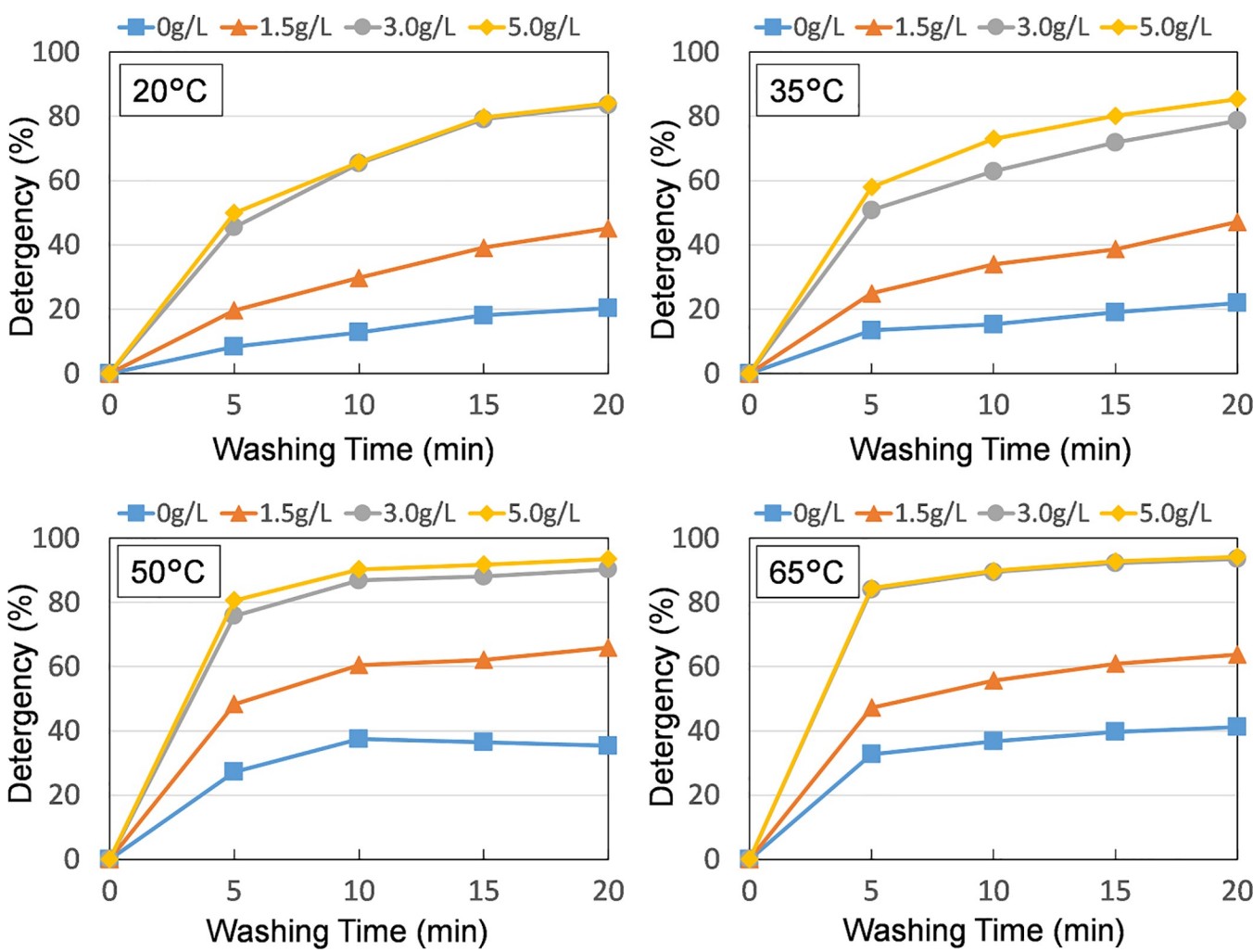

**Fig 6. Time-dependent removal curves of hemoglobin soil obtained with respect to various temperature and SDS concentration levels: Soiled fabrics were washed using a tergotometer (80 rpm).**

**Table 2. Calculated values of $\sigma_{rl}$ and $\mu_{rl}$ obtained from cleaning test of the hemoglobin-soiled cloth with respect to various temperature and SDS concentration levels.**

| SDS | 20°C | | 35°C | | 50°C | | 65°C | |
|---|---|---|---|---|---|---|---|---|
| | $\sigma_{rl}$ | $\mu_{rl}$ | $\sigma_{rl}$ | $\mu_{rl}$ | $\sigma_{rl}$ | $\mu_{rl}$ | $\sigma_{rl}$ | $\mu_{rl}$ |
| 0 g/L | 0.88 | -1.87 | 0.39 | -1.24 | 0.24 | -0.57 | 0.24 | -0.47 |
| 1.5 g/L | 1.16 | -1.35 | 0.73 | -0.88 | 0.45 | -0.03 | 0.44 | -0.09 |
| 3.0 g/L | 1.80 | -0.25 | 0.84 | -0.01 | 0.58 | +0.83 | 0.47 | +1.09 |
| 5.0 g/L | 1.38 | -0.05 | 0.93 | +0.26 | 0.63 | +1.03 | 0.49 | +1.12 |

increase in the surfactant concentration, and the surfactant was assumed to contribute to the dissolving action. In particular, in the case of washing using water that contained no SDS in the temperature range of 50–65°C, $\sigma_{rl}$ indicated that a mechanism for the removal of solid particulate soil can be realized. With the exception of initially removed stains, the remainder of the stains were difficult to remove thereafter.

The plot of $\mu_{rl}$ with respect to $1/T$ is shown in **Fig 7**. The cleaning tests were conducted in four steps of 20°C, 35°C, 50°C, and 65°C, and the cleaning performance at 65°C was slightly

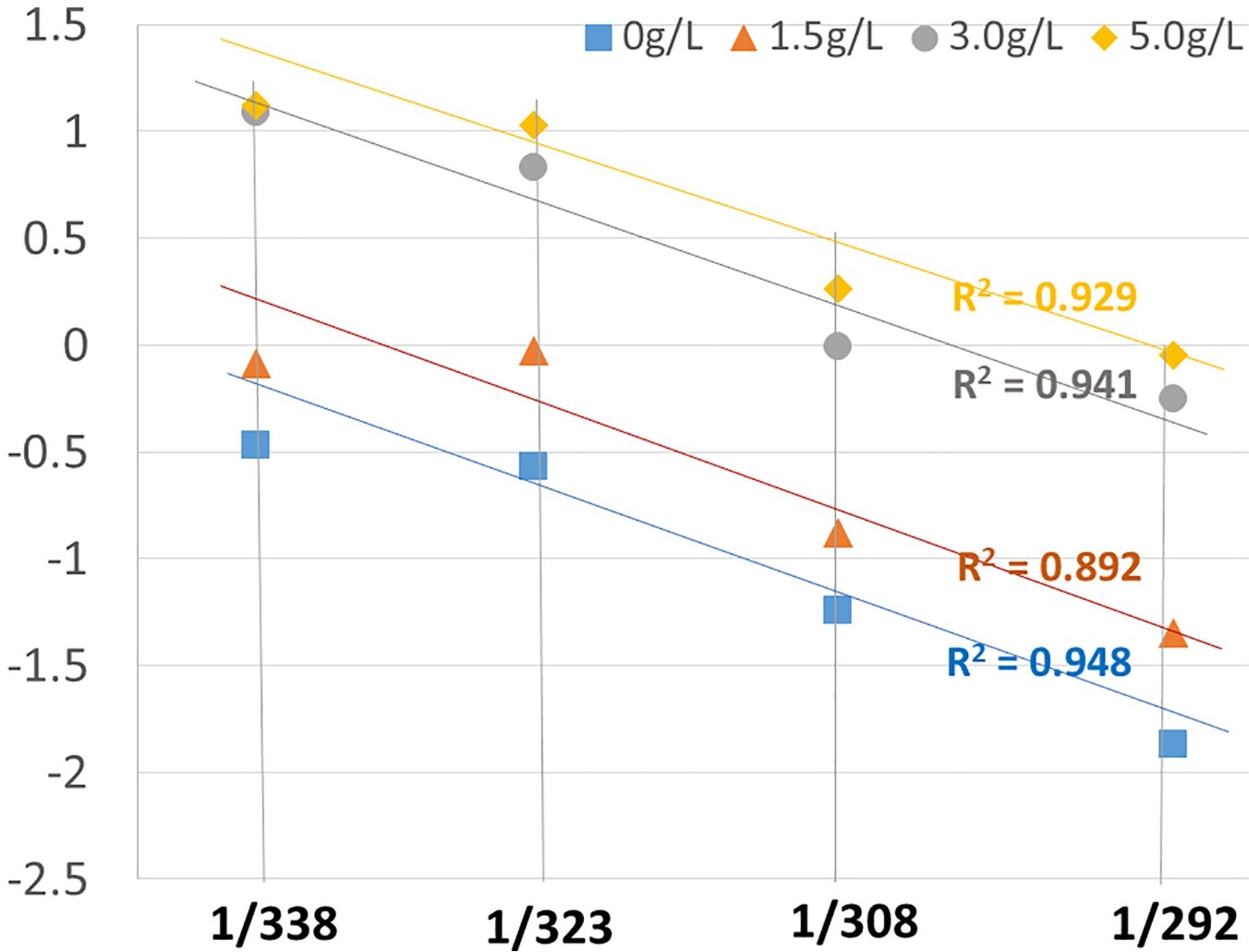

**Fig 7. Relationship between $\mu_{rl}$ and $1/T$ obtained from the washing test with respect to various SDS concentration and temperature levels.**

lower than that indicated by the $1/T$ plot. This can be attributed to the lack of increase in the washing efficiency, given that the protein denaturation temperature was approached.

### Verification of addition rule for surfactant concentration effect and temperature effect

In general, an increase in the SDS concentration and temperature result in an increase in the cleaning rate. Therefore, the interaction between the effects of increasing the SDS concentration and temperature was examined using the same evaluation method as that applied to the pH effect-temperature effect (Fig 8). Hence, the calculated and the experimental values were relatively close under the conditions of 20°C → 35°C and 35°C → 50°C

Moreover, in each of the three cases where the SDS concentration was changed with respect to a temperature change from 50°C to 65°C, a significant difference was observed between the

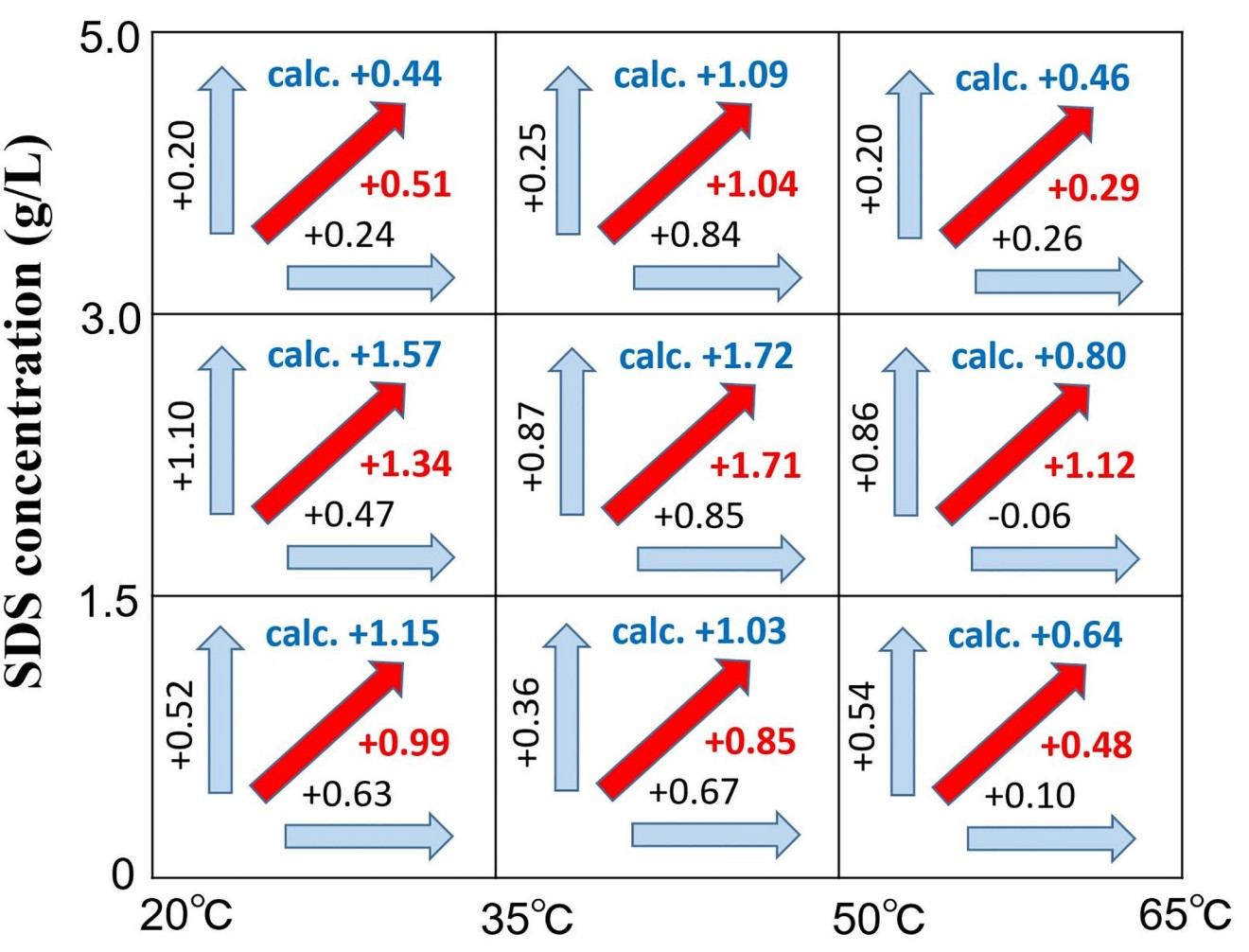

**Fig 8. Determination of additive rule between increase in SDS concentration and temperature in hemoglobin cleaning: The increase in $\mu_{rl}$ due to an increase in the SDS concentration is expressed as a vertical value, and that due to an increase in the temperature is expressed as a horizontal value.** The increase in $\mu_{rl}$ due to an increase in the SDS concentration and temperature is expressed as a diagonal arrow. Moreover, additivity is confirmed if the calculated value (vertical value + horizontal value) is close to the corresponding experimental value (value of red diagonal arrow).

calculated and experimental values. Under the conditions of [SDS concentration: 0 g/L → 1.5 g/L, 50˚C → 65˚C] and [SDS concentration: 3.0 g/L → 5.0 g / L, 50˚C → 65˚C], the experimental value was smaller than the calculated value. The expected concentration effect was not realized, given that the protein was denatured when the temperature reached 65˚C.

Conversely, under the conditions of [SDS concentration: 1.5 → 3.0 g/L, 50˚C → 65˚C], the experimental value was larger than the calculated value. It is a region of concentration change just toward cmc, in which case the dissolving power by SDS may have shown a temperature effect. Alternatively, the temperature range may have been within that of protein denaturation, which may have resulted in variations in the experimental values.

Thus, it is difficult to determine whether the addition rule is satisfied within the temperature range wherein protein denaturation occurs. However, within the range of 20–50˚C, an additive rule was observed between the effects of the SDS concentration and temperature.

## Conclusions

Based on the application of the probability density functional method to the analysis of the detergency of hemoglobin stains, additive effects were observed between the pH and temperature effects, and between the SDS concentration and temperature effects, under multiple conditions. Moreover, $\mu_{rl}$ was plotted linearly with respect to $1/T$ in both cases. Hence, $\mu_{rl}$ in this study can be employed as a parameter with properties similar to the rate constant. It is common knowledge that protein stains are denatured by external stimuli such as oxidation and high temperatures, which leads to removal difficulties. However, using the probability density functional method presented in this study has resulted in a link between cleaning rate and cleaning mechanism, which is rare in the field of cleaning. In particular, its potential as an effective tool for estimating compound effects was found. In the future, from the viewpoint that proteases weaken the adhesion of protein stains, it is thought that the research can be developed to quantify the effect on cleaning performance.

## Supporting information

**S1 Data.**
(DOCX)

**S2 Data.**
(DOCX)

## Author Contributions

**Conceptualization:** Miyako Oya, Masaru Oya.

**Data curation:** Yosuke Taniguchi.

**Investigation:** Yosuke Taniguchi, Naoaki Fujimura.

**Methodology:** Miyako Oya.

**Software:** Karen Miyamoto.

**Supervision:** Miyako Oya, Masaru Oya.

**Validation:** Masaru Oya.

**Writing – original draft:** Miyako Oya.

**Writing – review & editing:** Masaru Oya.

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
