## [Decision Letter · Decision Letter 0]

5 Jun 2020

PONE-D-20-08400

Kinetic analysis of hemoglobin detergency by Probability Density Functional Method

PLOS ONE

Dear Dr. Oya,

Thank you for submitting your manuscript to PLOS ONE. After careful consideration, we feel that it has merit but does not fully meet PLOS ONE’s publication criteria as it currently stands. Therefore, we invite you to submit a revised version of the manuscript that addresses the points raised during the review process.

We look forward to receiving your revised manuscript.

Kind regards,

Anjani Kumar Tiwari, Ph.D.

Academic Editor

PLOS ONE

Journal Requirements:

Additional Editor Comments (if provided):

The paper describes about the Kinetic analysis of hemoglobin detergency by Probability Density Functional Method. The concept seems very interesting but

Methodology is being required in more details for the sake of reproducibility of data. On what basis the group has decided to study only selective parameters (The process of cleaning has been studied as a function of pH, temperature and surfactant concentration), it should be explicitly explained in the manuscript.

The statistical parameter should also be mentioned in appropriate manner.

Reviewers' comments:

Reviewer's Responses to Questions

**Comments to the Author**

1. Is the manuscript technically sound, and do the data support the conclusions?

Reviewer #1: Yes

Reviewer #2: Yes

Reviewer #3: Partly

2. Has the statistical analysis been performed appropriately and rigorously? 

Reviewer #1: Yes

Reviewer #2: Yes

Reviewer #3: Yes

3. Have the authors made all data underlying the findings in their manuscript fully available?

Reviewer #1: Yes

Reviewer #2: Yes

Reviewer #3: Yes

4. Is the manuscript presented in an intelligible fashion and written in standard English?

Reviewer #1: No

Reviewer #2: Yes

Reviewer #3: No

5. Review Comments to the Author

Reviewer #1: Methodology is required for details for the sake of reproducibility of data. the concept is good and interesting but how do you select different parameters for detergency. this requires more information. the structuring of manuscript is a problem and require more changes in the draft.

Reviewer #2: In this manuscript, the authors report washing tests performed on medical samples prepared by soiling of fabric substrate with haemoglobin. There is sufficient data provided for the method of washing of samples with special reference to the calculation of two parameters αrl (Related to the cleaning mechanism) and µrl (Related to cleaning power).The process of cleaning has been studied as a function of pH, temperature and surfactant concentration and the data so obtained are explained on the basis of probability density functional method. The manuscript is well written and the results are explained properly.I recommend the publication of the manuscript in PLOSONE provided the following minor comments are addressed.

1. In the abstract: The first two lines in the beginning “Protein contamination----------theoron” should be deleted.

2 .In the abstract, the sentence starting from “ In this study------should be reworded and include washing tests performed by soiling of fabric and the continue as it is”

3.In the introduction: The lines 55 and 56 both starts both from “In this study..It should be corrected and reworded by deleting the words “in this” from line 55.

4. Materials and methods: Authors should mention the company and make of the thermostat/ water bath used for temperature control after pH in this section.

5.In Fig4,5: The authors should mention the values of regression coefficients(R2) either in the linear plots given in Fig 4 and Fig 5 or within the text at appropriate places.

Reviewer #3: The manuscript entitled "Kinetic analysis of hemoglobin detergency by Probability Density Functional Method " by

Miyako Oya and others seems very interesting but i have few observations which should be clarified for the researchers who wants to work in this direction

1- The author does not describe the nature of the cloth which is used and explained as

Approximately 250 g of cotton cloth was immersed in 2 L of 0.5% aqueous sodium carbonate

75 solution at 60–70 °C. The cloth was then rinsed using distilled water, dehydrated, air-dried, and cut it

into sections with dimensions of 5 × 5 cm2 76 .

2- Author should also explain that why specific pH and temperature variation were taken

why not pH 9 and temp 20 or 25°C

The time-dependent curves of the detergency obtained by varying the pH in steps of 6, 9, 10, and

11 at cleaning temperatures of 30 °C, 40 °C, 50 °C, and 60 °C

6. PLOS authors have the option to publish the peer review history of their article (what does this mean?). If published, this will include your full peer review and any attached files.

Reviewer #1: No

Reviewer #2: No

Reviewer #3: No

---

## [Author Response · Author response to Decision Letter 0]

26 Jun 2020

<To Editor>

Thank you for your valuable suggestions. It was corrected according to the instructions.

1) The following sentence was added to the introduction to explain the reason for determining the cleaning conditions.

(Revised Manuscript L66-70)

Since an alkaline solution is often used as a detergent for protein stains, it can be expected that the washing rate will change significantly depending on pH. Further, since SDS has a hemolytic action, it is expected that the concentration of SDS greatly contributes to the removability of hemoglobin stains. Moreover, since both can be regarded as a kind of chemical reaction, it can be predicted that they are affected by temperature.

2) The description of the statistical parameters was added as follows.

(Revised Manuscript L59-60)

the mean µrl and standard deviation σrl of the removal load distribution, assuming that the cleaning power follows a normal distribution. 

<To Reviewer #1>

Thank you for your valuable suggestions about the selection of parameters. The manuscript was corrected as follows.

(Revised Manuscript L66-70)

Since an alkaline solution is often used as a detergent for protein stains, it can be expected that the washing rate will change significantly depending on pH. Further, since SDS has a hemolytic action, it is expected that the concentration of SDS greatly contributes to the removability of hemoglobin stains. Moreover, since both can be regarded as a kind of chemical reaction, it can be predicted that they are affected by temperature.

The structure of the manuscript has already been corrected by Editage, but if you point out more specific problems, I would like to ask Editage to respond again.

<To Reviewer #2>

We would like to thank you for your careful review and for your valuable suggestions.

1. In the abstract: The first two lines in the beginning “Protein contamination----------theoron” should be deleted.

The manuscript has been revised as instructed.

2 .In the abstract, the sentence starting from “ In this study------should be reworded and include washing tests performed by soiling of fabric and the continue as it is”

The manuscript has been revised as follows.

(Revised Manuscript L3-4)

In this study, washing tests were performed using samples prepared by contaminating fabrics with hemoglobin, and a kinetic analysis

3.In the introduction: The lines 55 and 56 both starts both from “In this study..It should be corrected and reworded by deleting the words “in this” from line 55.

The manuscript has been revised as instructed.

4. Materials and methods: Authors should mention the company and make of the thermostat/ water bath used for temperature control after pH in this section.

The following explanationwas added in L80-81

(temperature was controlled in the washing machine)

5.In Fig4,5: The authors should mention the values of regression coefficients(R2) either in the linear plots given in Fig 4 and Fig 5 or within the text at appropriate places.

The regression coefficients were added in Fig.4 and Fig. 7.

<To Reviewer #3>

Thank you for your valuable suggestions. The following modifications were made according to your instructions.

1) An explanation about the specifications of the fabric was added as follows.

(Revised Manuscript L78)

 (weave density: 22.5/10 mm warp, 24.0/10mm weft, weight: 15.2 mg/cm2)

2) The following explanation was added about the reason why the test was performed at a temperature of 30°C or higher.

(Revised Manuscript L166-169)

Since it was expected that the cleaning efficiency would be low when the pH was low at low temperature and the cleaning effect due to the temperature and pH increase would be difficult to appear, the minimum temperature was set to 30 °C.

---

## [Decision Letter · Decision Letter 1]

23 Jul 2020

Kinetic analysis of hemoglobin detergency by Probability Density Functional Method

PONE-D-20-08400R1

Dear Dr. Oya,

We’re pleased to inform you that your manuscript has been judged scientifically suitable for publication and will be formally accepted for publication once it meets all outstanding technical requirements.

Kind regards,

Anjani Kumar Tiwari, Ph.D.

Academic Editor

PLOS ONE

Additional Editor Comments (optional):

Dear Dr Masaru Oya,

It is a pleasure to accept your manuscript entitled "Kinetic analysis of hemoglobin detergency by Probability Density Functional Method" in its current form for publication in PLOS ONE. The comments of the reviewer(s) who reviewed your manuscript are included at the foot of this letter.

Reviewers' comments:

Reviewer's Responses to Questions

**Comments to the Author**

1. If the authors have adequately addressed your comments raised in a previous round of review and you feel that this manuscript is now acceptable for publication, you may indicate that here to bypass the “Comments to the Author” section, enter your conflict of interest statement in the “Confidential to Editor” section, and submit your "Accept" recommendation.

Reviewer #1: All comments have been addressed

Reviewer #3: All comments have been addressed

2. Is the manuscript technically sound, and do the data support the conclusions?

Reviewer #1: Yes

Reviewer #3: Yes

3. Has the statistical analysis been performed appropriately and rigorously? 

Reviewer #1: Yes

Reviewer #3: Yes

4. Have the authors made all data underlying the findings in their manuscript fully available?

Reviewer #1: Yes

Reviewer #3: Yes

5. Is the manuscript presented in an intelligible fashion and written in standard English?

Reviewer #1: Yes

Reviewer #3: Yes

6. Review Comments to the Author

Reviewer #1: the revised manuscript has addressed all the necessary clarifications and I expressed my satisfaction with the current status of the revised manuscript.

Reviewer #3: The authors of manuscript responed reviewers query in well manner and now it can be considered for publication.

7. PLOS authors have the option to publish the peer review history of their article (what does this mean?). If published, this will include your full peer review and any attached files.

Reviewer #1: No

Reviewer #3: No

---

## [Editor Report · Acceptance letter]

29 Jul 2020

PONE-D-20-08400R1 

Kinetic analysis of hemoglobin detergency by Probability Density Functional Method 

Dear Dr. Oya:

I'm pleased to inform you that your manuscript has been deemed suitable for publication in PLOS ONE. Congratulations! Your manuscript is now with our production department. 

Kind regards, 

on behalf of

Dr. Anjani Kumar Tiwari 

Academic Editor

PLOS ONE